# Myxolipoma of the Popliteal Fossa: A Rare Tumor Case Report

**DOI:** 10.3390/reports7030058

**Published:** 2024-07-25

**Authors:** Yuchen You, Jessica Cao, Brandon Nguyen, Melanie Gero, Karim Jreije

**Affiliations:** 1Ventura County Medical Center, Ventura, CA 93003, USA; yyou1@cmhshealth.org; 2Community Memorial Hospital, Ventura, CA 93003, USA; jcao1@cmhshealth.org (J.C.); bnguyen2@cmhshealth.org (B.N.); melanie.gero@ventura.org (M.G.)

**Keywords:** myxolipoma, popliteal fossa, rare tumor, case report

## Abstract

Myxolipomas are rare variants of lipomas characterized by abundant myxoid changes resulting from an abundant mucoid component. While myxolipomas have been reported in various anatomical locations, their occurrence in the popliteal fossa is exceptionally rare, with the last published case dating back to 1914. We present a case of a 64-year-old male with a large myxolipoma in the popliteal region. The patient underwent successful surgical excision, and a histopathological examination confirmed the diagnosis of myxolipoma. This case report highlights the clinical features, differential diagnosis, and diagnostic challenges associated with myxolipomas in the popliteal fossa.

## 1. Introduction

Lipomas, commonly encountered benign neoplasms derived from mature adipocytes, often present as soft, painless masses. However, within the realm of lipomatous tumors, a spectrum of histological variants exists, each displaying distinctive mesenchymal characteristics. Among these variants, myxolipomas emerge as rare entities distinguished by pronounced myxoid alterations resulting from an abundance of mucoid elements. While myxolipomas have been sporadically documented in various anatomical locations such as the heart, tongue, oral cavity, and retroperitoneal region, their occurrence within the popliteal fossa remains exceptionally scarce, with historical records dating back more than a century [1,2]. In our article, we aim to present and dissect a recent case of myxolipoma localized to the popliteal region, elucidating its clinical manifestation, diagnostic approach, therapeutic interventions, and histopathological characteristics within the context of a county hospital setting. Through meticulous examination of this unique clinical scenario, we endeavor to enrich the medical literature with valuable insights, thereby enhancing awareness regarding the potential manifestation of myxolipomas in the popliteal fossa. Furthermore, we seek to delve into the nuances that distinguish myxolipomas from other benign lipomatous tumors, emphasizing the imperative of accurate discrimination from myxoid liposarcomas, a malignant neoplasm sharing histological similarities with myxolipomas. In navigating this intricate diagnostic landscape, clinicians must exercise keen discernment to ensure precise identification and tailored management strategies for patients afflicted with lipomatous tumors. By fostering a comprehensive understanding of the subtle histopathological variances and clinical implications inherent to these entities, healthcare providers can optimize patient care, refine treatment algorithms, and ultimately improve therapeutic outcomes and prognostic outlooks for affected individuals.

## 2. Detailed Case Description

A 64-year-old male with a history of hypertension and heavy alcohol consumption was referred to us by his primary provider with a longstanding mass in the right popliteal region. The patient reported an absence of pain and no restriction in their range of motion, although he observed a gradual increase in mass size over six years. The initial examination six years ago revealed a painless, small soft tissue mass in the popliteal fossa. Further assessment at that time employing ultrasound confirmed a mobile soft tissue mass measuring 4.2 cm × 2.4 cm × 3.6 cm, subsequently confirmed via core needle biopsy as a lipoma. Close patient monitoring over the ensuing six years revealed a nearly twofold increase in mass dimensions. Therefore, a contrasted MRI was performed, which exhibited homogeneous, low signal intensity on a T1-weighted pulse sequence and heterogenous, high signal intensity on T2-weighted pulse sequence for the mass within the popliteal fossa, measuring 8.0 cm × 7.0 cm × 5.0 cm (Figure 1). The mass was subcutaneous and located proximate to the rectus femoris and distal aspect of the vastus lateralis muscle. Areas of septations were observed within the mass, raising concerns regarding the possibility of liposarcoma.

Given the mass’s substantial size and anatomical location, the patient was referred for surgical evaluation. Surgical examination unveiled a soft, mobile, and non-tender mass measuring approximately 8 cm, devoid of local lymphadenopathy or additional masses.

Complete excision of the mass was planned and successfully performed under general anesthesia without complications. The mass displayed clear demarcation and facile separation from the underlying neurovascular bundle. The excised mass exhibited a lobulated, soft, whitish-yellow appearance (Figure 2 and Figure 3). Histopathological analysis disclosed an encapsulated and lobulated mass of mature adipose tissue exhibiting prominent stromal myxoid changes. It was positively stained with alcian blue and was digestible by hyaluronidase. The mass measured 10 × 9.5 × 3 cm on final pathology, with no signs of cellular atypia, lipoblasts, chicken wire capillary networks, or necrosis. Immunohistochemical studies, including FISH investigations for MDM2 and DDIT3, yielded negative results. (Figure 4) Consequently, a definitive diagnosis of myxolipoma was established. The postoperative period was uneventful, and the patient recovered well without major complications upon re-evaluation two weeks after the surgery.

## 3. Discussion

Lipomas predominantly occur in the trunk and proximal extremities, with a notably rare occurrence in the popliteal fossa and lower leg [3]. Moreover, the predominant mass often encountered in the popliteal fossa is the popliteal cyst, also known as a Baker’s cyst [4]. Akiyama’s study of 86 surgically resected lipomas found approximately 8% to be located in the lower leg [5].

Myxoid lipomas represent a distinctive subtype of lipomatous tumor histologically characterized by the presence of adipocytes surrounded by myxoid material, which imparts a characteristic myxomatous appearance (Table 1). The identification of glycosaminoglycans within the myxoid matrix can be facilitated through specialized stains such as alcian blue, aiding in their histological confirmation [6]. Lipomas are often diagnosed clinically and sent for histologic evaluation after surgical excision. However, there are certain factors for which radiologic imaging of these tumors is beneficial, such as rapid growth, pain, atypical presentation, and deep location [7]. In clinical practice, magnetic resonance imaging (MRI) is preferred for radiologic assessment due to its superior capability in visualizing soft tissues. Conversely, demarcating myxolipomas on computed tomography (CT) can be challenging, particularly in the presence of surrounding adipose tissue. However, they may manifest on CT imaging as heterogeneous solid masses with nodular growth. In contrast, MRI typically reveals myxolipomas as exhibiting homogeneous, low signal intensity on T1-weighted images and heterogeneous, high signal intensity on T2-weighted images [8]. Immunohistochemically, myxolipomas remain underexplored and would greatly benefit from further research (Table 1).

Distinguishing myxolipoma from other benign lipomatous tumors, such as chondroid lipoma and spindle cell lipoma, necessitates a nuanced understanding of their microscopic characteristics and clinical features.

Chondroid lipomas, for instance, are rare, benign adipocytic tumors that predominantly affect adult females in the extremities and limb girdles [9,10]. The tumor is usually painless and slow-growing. Diagnosis is not easily made with CT due to the lack of defining characteristics. On MRI, chondroid lipomas typically exhibit variable T1 signals, with the majority displaying high T2 signals. Notably, Yildez et al. observed a distinctive “fat ring” on contrast-enhanced, T2-weighted MRI [11]. Histologically, they are characterized by the presence of nests and cords of rounded cells with granular eosinophilic or multivacuolated, lipid-containing cytoplasm within a prominent myxohyaline stroma [9] (Table 1). Cell features include cytoplasmic vacuolation, eosinophilic granular cytoplasm resembling lipoblasts, and oval-to-reniform nuclei with occasional irregularities. The stroma exhibits myxoid areas interspersed with foci of hyalinization, accompanied by hemorrhage, sclerosis, and vascular abnormalities, occasionally including inflammation, calcification, or metaplastic bone formation [12].

Chondroid lipomas display varied immunohistochemical positivity for vimentin, S-100 protein, and CD68, with occasional cytokeratin expression and high cyclin D1 expression (Table 1). They are negative for HMB45, smooth muscle actin, muscle-specific actin, glial fibrillary acidic protein, and Leu-7 (CD57). Periodic acid Schiff and reticulin are positive around small cells, indicating a basal lamina, while periodic acid Schiff positivity in vacuolated eosinophilic cells suggests cytoplasmic glycogen; alcian blue and colloidal iron stains were positive, indicating extracellular acid mucopolysaccharides with variable sensitivity to hyaluronidase digestion [13].

Cytogenetically, chondroid lipomas consistently exhibit reciprocal translocations t (11;16) (q13;p13), resulting in the C11orf95-MKL2 fusion oncogene [14] (Table 1). Chondroid lipoma is often benign and there are no reported cases of local recurrence after complete surgical excision [12].

Chondroid lipomas can be confused with spindle cell lipomas, which primarily occur on the back or posterior neck, display CD34 expression, and feature deletions involving 13q and 16q. Distinction from well-differentiated liposarcomas is important as they usually affect older individuals and lack an extensive myxochondroid matrix, while myxoid liposarcomas, affecting younger individuals, display a delicate branching capillary network and characteristic chromosomal translocations. Extraskeletal myxoid chondrosarcoma, typically occurring in older males, lacks a fatty component and shows recurrent chromosomal translocations involving NR4A3. Myoepithelial tumors, often superficial, exhibit cytokeratin expression and various genetic rearrangements, differing from chondroid lipomas [12].

Spindle cell lipomas (SCLs) typically arise in the posterior neck and upper back of elderly males [15]. Histologically SCLs consist of mature adipocytic components accompanied by bland spindle cells, ropey collagen fibers, mast cells, and areas of myxoid degeneration [16] (Table 1). Immunohistochemically, SCLs stain positive for CD34 and negative with S-100 protein [17] (Table 1). Myxoid variants of SCLs typically exhibit a prominent myxoid stroma primarily composed of hyaluronic acid [18]. On MRI, SCLs can appear low in fat, similar to the more aggressive myxoid liposarcoma [19]. In ambiguous cases, molecular testing such as fluorescence in situ hybridization (FISH) can be valuable for detecting RB1 gene deletion in spindle cell lipomas [20]. 

Angiolipoma is a benign tumor that usually arises in the subcutaneous torso, extremities, and neck [21]. Surgical features may show fatty tissue, with port wine or dark brown discoloration. Angiolipomas are often well circumscribed and encapsulated and small (<2 cm) [21]. Histologically, they are distinguished by mature fat cells with peripheral capillary vessels and fibrin thrombi [21] (Table 1). Immunohistochemical stains involve S-100 fat cells and CD31/34 endothelium [21] (Table 1). MRI demonstrates hypointense vascular elements and hyperintense fat components on T1 imaging and hyperintense vascular elements on T2 [21]. Angiolipomas that are found intramuscularly are hemangiomas [21].

Fibrolipoma is another variant of lipoma that is commonly found in the buccal mucosa, tongue, and floor of the mouth [22]. Surgically it is characterized by a significant fibrous component surrounding the fat cells, with the consistency ranging from soft to firm depending on the fibrous element [22]. On MRI, they demonstrate well-encapsulated masses with high intensity on both T1- and T2-weighted imaging [23]. Immunohistochemistry shows vimentin positivity and Ki-67 antibody expression [23] (Table 1). The histological description of fibrolipoma reveals underlying fibrous connective tissue containing bundles of collagen fibers interspersed with lobules of adipocytes [24] (Table 1).

Accurate differentiation between myxolipomas and myxoid liposarcomas is crucial since the latter represents a malignant neoplasm and requires different treatment and management. Biopsy-proven myxoid liposarcomas are treated with wide local excisions with 2 cm margins, with consideration of neoadjuvant chemotherapy and/or radiotherapy based on tumor size or surrounding neurovascular structures [25]. In contrast to myxolipomas, liposarcomas are firm, adherent to surrounding tissues, and typically located in deeper soft tissues such as the lower extremities and retroperitoneum, predominantly affecting individuals later in life [26]. On MRI, myxoid liposarcomas are seen as a multilobulated, heterogeneous mass, often with septations. The myxoid component appears as hyperdense on T2 and hypodense on T1 and enhances with contrast. The fatty component is hyperintense on T1 and, frequently, is a low percentage of the entire tumor volume [27]. Histologically, myxoid liposarcomas predominantly consist of uniform round- to oval-shaped primitive mesenchymal cells mixed with signet rings or multivacuolated lipoblasts within a prominent myxoid stroma displaying a delicate “chicken-wire” capillary vasculature [28] (Table 1). Cytogenetically, myxoid liposarcomas are characterized by translocations t (12;16)(q13;p11), leading to the formation of a FUS-DDIT3 fusion gene [29]. Immunohistochemically, myxoliposarcomas can be differentiated by DDIT3 reactivity [30]. 

## 4. Conclusions

In conclusion, myxolipomas can be differentiated from other benign and malignant lipomatous tumors based on specific histological characteristics and clinical features. MRI can be helpful but is often nonspecific. In ambiguous cases, immunohistochemical testing can distinguish between benign and malignant lipoma subtypes. Understanding the differences between these tumors is essential for appropriate diagnosis and management. 

This case report was written using the SCARE guidelines [31]. Written informed consent was obtained from the patient to publish this case report and accompanying images. 

## Figures and Tables

**Figure 1 reports-07-00058-f001:**
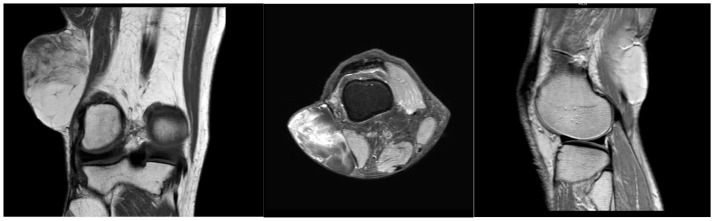
MRI coronal, transverse, and sagittal view of right popliteal fossa mass.

**Figure 2 reports-07-00058-f002:**
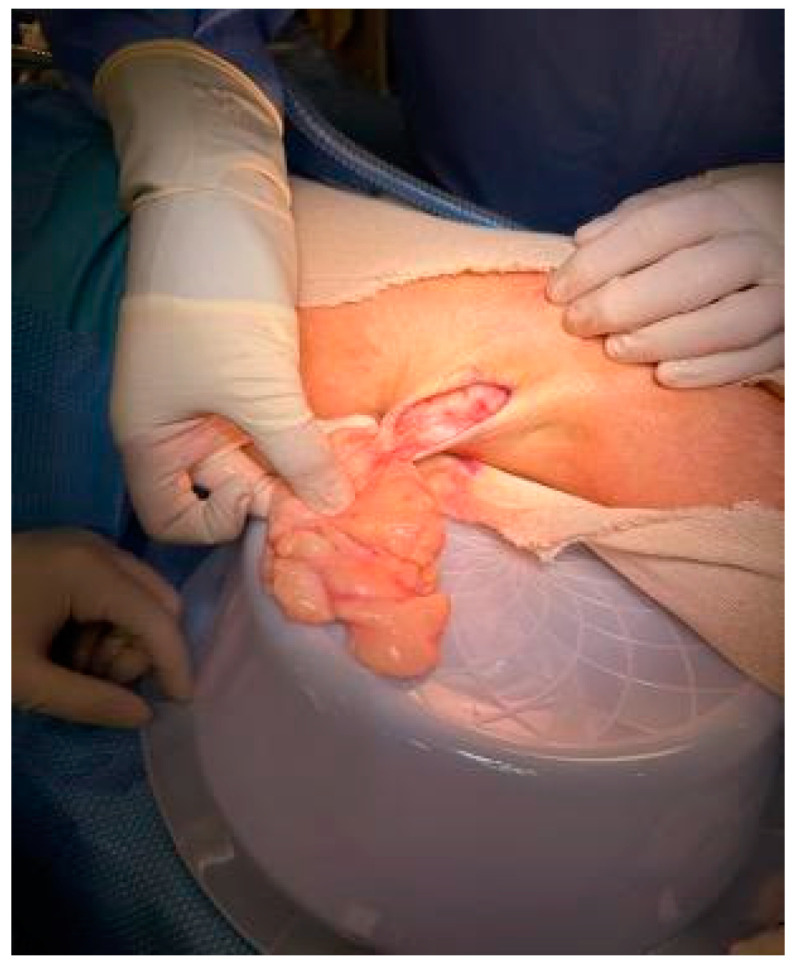
Macroscopic appearance of excised mass.

**Figure 3 reports-07-00058-f003:**
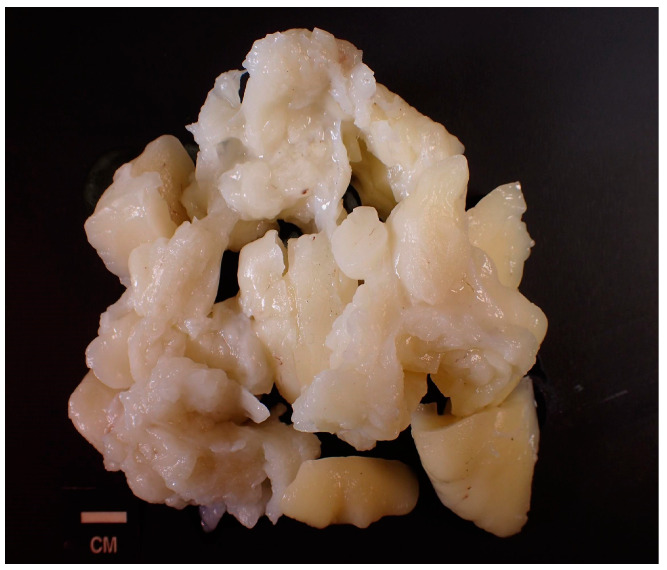
Isolated view of excised mass exhibiting a lobulated, soft, whitish-yellow appearance.

**Figure 4 reports-07-00058-f004:**
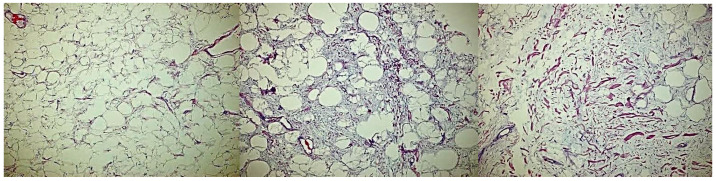
Microscopic views of the mass shows no signs of cellular atypia, lipoblasts, chicken wire capillary networks, or necrosis.

**Table 1 reports-07-00058-t001:** Differentiation of the subtypes of lipomas.

	Myxoid Lipomas	Chondroid Lipoma	Spindle Cell Lipoma	Myxoid Liposarcoma	Angiolipoma	Fibrolipoma
**Histology**	Adipocytes surrounded by myxoid material	Embryonal fat and cartilage nests; myxohyaline stroma	Mature adipocytes accompanied by spindle cells, collagen fibers, mast cells, and areas of myxoid degeneration	Primitive mesenchymal cells mixed with signet rings or multivacuolated lipoblasts within “chicken-wire” capillary vasculature	Composed of mature fat with numerous small blood vessels, with fibrin thrombi present	Bundles of collagen fibers interspersed with lobules of adipocytes.
**MRI**	T1: homogeneous, low-intensity signal T2: Heterogeneous, high-intensity signal	T1: variable signals; T2: high signals. “Fat ring” on contrast-enhanced T2W MRI	Can appear low in fat, similar to the more aggressive myxoid liposarcoma	Multilobulated, heterogeneous mass T1: hypodense myxoid; enhances with contrast. Hyperdense fatty component (low percentage of tumor) T2: hyperdense myxoid component	T1: hypointense vascular elements; hyperintense fatty components T2: hyperintense vascular elements	High-intensity and well-encapsulated on both T1 and T2 imaging
**IHC**	-	Vimentin, S100, and CD68	CD34	DDIT3	S100, CD31/34/61	Vimentin, Ki-67

## Data Availability

The data presented in this study are available on request from the corresponding author due to patient privacy.

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
