# Peer review of "Myxolipoma of the Popliteal Fossa: A Rare Tumor Case Report"

_reports, 2024, doi:10.3390/reports7030058_

Round 1

Reviewer 1 Report

Comments and Suggestions for Authors

The case of a rare case of lipoma of the popliteal fossa, surgically resected, is presented.

It deserves publication because it is a rare variety of lipoma.

The article is well written, the spelling and syntax is adequate.

I only suggest that the quality of the microscopic photography of the piece be improved, it is very poor.

It can be published if what is indicated is corrected.

Author Response

Thank you for your valuable review of our case report; we greatly appreciate it. We sincerely appreciate you taking the time to provide us with feedback. We have carefully considered your insights and made extensive improvements to our paper accordingly. Regarding your concerns of the microscopic photography, we have significantly optimized the images to improve their clarity and interpretability in the new document.

Reviewer 2 Report

Comments and Suggestions for Authors

- The introduction lacks sources, kindly refer to the related references 

- In case presentation: Authors mentioned that that the preoperative diagnosis was done (via biopsy), what type of biopsy??

- Authors claim that the immunohistochemistry study includes FISH , however FISH and genetic studies are ancillary testing

- Details of the immunohistochemistry markers used are essential, however authors did not

- Histological differential diagnosis is crucial, kindly mention in case presentation and the discussion 

Comments on the Quality of English Language

OK

Author Response

Thank you for your valuable review of our case report; we greatly appreciate it. We sincerely appreciate you taking the time to provide us with feedback. We have carefully considered your insights and made extensive improvements to our paper accordingly.

Regarding the biopsy type, the preoperative diagnosis was done via core needle biopsy. We will include this into our manuscript.

We have incorporated additional immunohistochemistry details into the paper while removing specifics related to FISH.

We have edited the report with additional details concerning histological differential diagnosis.

Thank you very much for your time.

Reviewer 3 Report

Comments and Suggestions for Authors

Dear Editor,

Thank you for giving me an opportunity to review the manuscript about myxoid lipoma in popliteal fossa, a case report

I would like the authors to address the following points I pointed out.

1.      Actual MRI image is required.

2.      If the authors mention the rarity of location, literature review about tumor location should be added.

Author Response

Thank you for your valuable review of our case report; we greatly appreciate it. We sincerely appreciate you taking the time to provide us with feedback. We have carefully considered your insights and made extensive improvements to our paper accordingly.

We have added an MRI image to our study. Additionally we have incorporated a literature review discussing the rarity of this type of tumor in the popliteal fossa into our study.

Reviewer 4 Report

Comments and Suggestions for Authors

The authors show the interesting case of myxolipoma of the popliteral fossa. I have some concerns to discuss.

1. What other diagnoses were considered in this case before arriving at the diagnosis of myxolipoma? What steps were taken to exclude those diagnoses? 2. How is the treatment of myxolipoma different from other lipoma subtypes, and why was that treatment chosen? 3. Did this patient experience recurrence or complications after surgery, and what are some common complications that patients with myxolipoma may experience? 4. Any interesting blood results? 5. How does this differ from intraosseous lipomas? Discuss whether it is more aggressive. With reference to the following literature: Aggressive intraosseous lipoma of the scapula: A case report. Exp Ther Med. 2023;26(2):400. Published 2023 Jul 6. doi:10.3892/etm.2023.12099

Author Response

Thank you for your valuable review of our case report; we greatly appreciate it. We sincerely appreciate you taking the time to provide us with feedback. We have carefully considered your insights and made extensive improvements to our paper accordingly.

  1. What other diagnoses were considered in this case before arriving at the diagnosis of myxolipoma? What steps were taken to exclude those diagnoses? 

The top differential diagnosis for this patient was a conventional lipoma. This is due to its subcutaneous, soft, and mobile nature.

  1. How is the treatment of myxolipoma different from other lipoma subtypes, and why was that treatment chosen? 

The treatment of myxolipoma is similar to the treatment to the conventional lipoma as it is still a benign tumor. A malignant liposarcoma would have a different treatment type consisting of wide local excision (with 2cm margins) with chemoradiation considerations.

  1. Did this patient experience recurrence or complications after surgery, and what are some common complications that patients with myxolipoma may experience? 

The patient did not experience any recurrance of complications after surgery. Common complications with similar patients consist of surgical site infection, hematoma, and recurrance.

  1. Any interesting blood results? 

All blood results for this patient was within normal limits.

  1. How does this differ from intraosseous lipomas? Discuss whether it is more aggressive. With reference to the following literature: Aggressive intraosseous lipoma of the scapula: A case report. Exp Ther Med. 2023;26(2):400. Published 2023 Jul 6. doi:10.3892/etm.2023.12099

Intraosseous lipomas describe the anatomical location of a lipoma inside of bone, while myxolipoma is a subtype of lipoma (our case describes a myxolipoma in the subcutaneous tissue of the popliteal fossa). They are both generally not aggressive as they are both benign tumors. The case listed above, including ours, required surgical intervention due to continued growth of the tumor.

Thank you for your time.

Round 2

Reviewer 3 Report

Comments and Suggestions for Authors

The authors addressed my concern

Reviewer 4 Report

Comments and Suggestions for Authors

The manuscript is suitable for publication.